# *Chelidonium**majus* L. Incorporated Emulsion Electrospun PCL/PVA_PEC Nanofibrous Meshes for Antibacterial Wound Dressing Applications

**DOI:** 10.3390/nano11071785

**Published:** 2021-07-09

**Authors:** Cláudia Mouro, Ana P. Gomes, Merja Ahonen, Raul Fangueiro, Isabel C. Gouveia

**Affiliations:** 1FibEnTech Research Unit, Faculty of Engineering, University of Beira Interior, 6201 001 Covilhã, Portugal; d1684@ubi.pt (C.M.); anapaula@ubi.pt (A.P.G.); 2Faculty of Technology and WANDER Nordic Water and Materials Institute, Satakunta University of Applied Sciences, 26101 Rauma, Finland; merja.ahonen@samk.fi; 3Centre for Textile Science and Technology (2C2T), University of Minho, 4800 058 Guimarães, Portugal; rfangueiro@dem.uminho.pt

**Keywords:** emulsion electrospinning, needleless electrospinning, natural compounds, *C. majus*, antimicrobial wound dressing

## Abstract

Presently, there are many different types of wound dressings available on the market. Nonetheless, there is still a great interest to improve the performance and efficiency of these materials. Concerning that, new dressing materials containing natural products, such as medicinal plants that protect the wound from infections but also enhance skin regeneration have been or are being developed. Herein, we used for the first time a needleless emulsion electrospinning technique for incorporating *Chelidonium*
*majus* L. (*C. majus*), a medicinal plant widely known for its traditional therapeutic properties, in Polycaprolactone (PCL)/Polyvinyl Alcohol (PVA)_Pectin (PEC) nanofibrous meshes. Moreover, the potential use of these electrospun nanofibers as a carrier for *C. majus* was also explored. The results obtained revealed that the produced PCL/PVA_PEC nanofibrous meshes containing *C. majus* extract displayed morphological characteristics similar to the natural extracellular matrix of the skin (ECM). Furthermore, the produced meshes showed beneficial properties to support the healing process. Additionally, the *C. majus*-loaded PCL/PVA_PEC nanofibrous meshes inhibited *Staphylococcus aureus* (*S. aureus*) and *Pseudomonas aeruginosa* (*P. aeruginosa*) growth, reaching a 3.82 Log reduction, and showed to be useful for controlled release, without causing any cytotoxic effect on the normal human dermal fibroblasts (NHDF) cells. Hence, these findings suggest the promising suitability of this novel wound dressing material for prevention and treatment of bacterial wound infections.

## 1. Introduction

Over recent decades, traditional wound dressings have been applied to protect the wound from mechanical and bacterial injuries, and simultaneously to absorb low levels of exudates. However, although these dressings are cheaper and provide an essential environment for wound healing, they are not efficient enough to promote hemostasis and maintain a moist wound environment, which has been shown to enhance wound healing [1,2,3,4,5,6]. To overcome such limitations and improve the healing process, different modern wound dressings such as films, foams, hydrogels, hydrocolloids, and micro to nanofibers meshes have been fabricated from a wide range of natural and synthetic biomaterials and selected based on the type of wounds [4,5,6,7]. Among them, nanofiber meshes produced from several methods, including self-assembly, phase separation, drawing, template synthesis, and electrospinning have been regarded as one of the most efficient wound dressing materials, especially those produced by electrospinning, due to its simplicity, cost-effectiveness, and functional versatility [1,2,4,6,8,9].

The electrospun nanofibrous meshes exhibit desirable features for wound healing occurs, once they form randomly orientated 3D-structures which resemble the skin’s extracellular matrix (ECM) [4,6,8,9,10,11]. Moreover, these meshes display high surface area, high porosity, and interconnected pores, which provide a suitable microenvironment for cell attachment, growth, and differentiation, as well as angiogenesis and collagen synthesis [3,4,6,8,9,12,13,14]. Furthermore, the electrospun nanofibers can control gas exchanges, supply water and nutrients to the wound site, and assist in fluids absorption, maintaining a moisture balance at the wound dressing interface [4,6,13]. These materials also exhibit potential to scar-free repair and regeneration, and the loading ability for a wide variety of bioactive or therapeutic agents, including antimicrobials, anti-inflammatories, growth factors, vitamins, and even cells [4,6,8,9,12,13,14,15,16].

The incorporation of these agents into electrospun nanofibrous meshes has been performed using different strategies, including blend, co-axial, and emulsion electrospinning. However, post-electrospinning surface modifications, such as chemical immobilization, coating, and physical absorption, as well as layer-by-layer assembly have also been considered [4,6,9,14,17,18].

Among the different methods used until now to add bioactive agents to electrospun nanofibers, emulsion electrospinning has gained considerable attention for successfully loading both hydrophilic and hydrophobic agents, and capability to protect the structural integrity and bioactivity of these agents [4,19]. Additionally, it does not require any specific setup and can enhance the solubility of the components of the blend, as well as act as delivery systems, improving its therapeutic performance [4,18,19,20]. In this method, an emulsion is electrospun into core-sheath structured fibers due to the difference in volatility between the two liquid phases, i.e., one liquid is dispersed into the other in the form of droplets, and during the electrospinning process, the solvent of the dispersed phase evaporates more rapidly while the emulsion droplets move inward and merge in the core of the fibers [18,20]. Moreover, the electrospun nanofibrous meshes prepared by emulsion electrospinning have shown high encapsulation efficiency [4].

Despite the efficient incorporation of a wide range of bioactive and therapeutic agents into electrospun nanofibers, the interest for natural compounds remains high [4,9,21]. Recent studies report the development of electrospun nanofibrous membranes containing medicinal plant extracts, and their constituents, such as essential oils for wound dressing applications due to their therapeutic properties, cost-effectiveness, and few adverse effects [4,9,21,22,23,24,25,26,27,28]. Plant-derived medicines have a long history of use in traditional medicine, namely in the treatment of different kinds of wounds [22,27,28]. Hence, the electrospun nanofibrous meshes containing plant extracts are a promising therapeutic option, and some practical examples were already successfully applied as an advantageous approach to prevent wound bacterial colonization and infection, while provide an appropriate environment for healing [4,6,9,21,22,23,24,25,26,27,28].

Herein, we aimed to fabricate electrospun Polycaprolactone (PCL)/Polyvinyl Alcohol (PVA)_Pectin (PEC) nanofibrous meshes containing crude *Chelidonium*
*majus L.* (*C. majus*) extract from emulsion electrospinning. *C. majus* is a member of the *Papaveraceae* family, and it is used for the treatment of different types of warts and various inflammatory diseases, including atopic dermatitis [29,30,31,32]. The crude *C. majus* extract, as well as purified compounds derived from it, exhibit a broad spectrum of biological properties. The flavonoids and isoquinoline alkaloids, such as chelidonine, berberine, coptisine, sanguinarine, and chelerythrine, are responsible for conferring anti-inflammatory, antimicrobial, anticancer, analgesic, antioxidant, and hepatoprotective properties to *C. majus* [29,30,31,32].

On the other hand, the hydrophobic character, strength, and durability of the synthetic polymer, PCL, was combined with the intrinsic biological properties displayed by PEC [29,33,34,35,36]. The PEC is a natural polysaccharide, known for its biocompatible, biodegradable, and non-toxicity. Additionally, this natural biopolymer can promote a favorable environment for cell adhesion and proliferation and provide an acid environment which may act as a barrier against bacteria [29,36,37]. Nonetheless, natural biopolymers, as polysaccharides are usually tricky to electrospun into nanofibers, and therefore PVA, one of the most frequently used synthetic water-soluble polymer for the preparation of wound dressings was added to improve PEC electro-spinnability [36]. As a result, PVA is an excellent emulsifying and dispersing agent and can facilitate emulsification and promote emulsion stability [38].

To the best of our knowledge, this is the first time that electrospun PCL/PVA_PEC nanofibrous meshes containing crude *C. majus* extract were successfully fabricated. The obtained results revealed the appropriateness of incorporating a medicinal plant into nanofiber-based wound dressings via emulsion electrospinning, aiming to prevent bacterial infection at the wound site and enhance the healing process.

## 2. Materials and Methods

### 2.1. Materials

*Chelidonium**majus* L. (*C. majus*) was bought from a Portuguese botanic shop. Normal human dermal fibroblasts (NHDF) cells were acquired from ATCC (American Type Culture Collection, Manassas, VA, USA). Polycaprolactone (PCL) (MW 80,000 g/mol) was purchased from Sigma-Aldrich (St. Louis, MO, USA). Polyvinyl Alcohol (PVA) (MW 115,000 g/mol) was obtained from VWR Chemicals (Geldenaaksebaan, Leuven, Belgium). Pectin (PEC) was purchased from Acros Organics (Fisher Scientific, Göteborg, Sweden). Chloroform (analytical grade), Dimethylformamide (DMF) (analytical grade), and ethanol absolute were acquired from Fisher Chemical (Leicestershire, UK). Nutrient agar (NA), Nutrient broth (NB), and agar for microbiology were bought from Fluka (Barcelona, Spain). Brain Heart Infusion (BHI) broth was purchased from Panreac (Barcelona, Spain). Sodium chloride (NaCl), Mueller-Hinton broth (MHB), tween 80, trypsin, fetal bovine serum (FBS), 3-(4,5-Dimethyl-2-thiazolyl)-2,5-diphenyl-2H-tetrazolium bromide (MTT), and Dimethyl Sulfoxide (DMSO) anhydrous ≥99.9% were provided from Sigma-Aldrich. Phosphate-buffered saline (PBS) was purchased from Alfa Aesar (Kandel, Germany). All solvents were used as received without further purification.

### 2.2. Preparation of Crude C. majus Extract

#### Ethanol Crude Extraction of *C. majus*

Dried plant material (2.5 g) was extracted using 80% ethanol at room temperature. After that, the plant extract was collected and filtered with filter paper (Whatman No. 1, 11 µm pore size), and the filtrate was then subjected to evaporation by rotavapor (Buchi Rotavapor RE 111, Allschwil, Switzerland) to obtain a soft extract. The yield of the extract was 11.29% (*w*/*w*). The crude *C. majus* extract was re-suspended in ethanol and used for all experiments, as described below.

### 2.3. In Vitro Minimum Inhibitory Concentration (MIC) Test of the Crude Extract of C. majus

Minimum inhibitory concentration (MIC) of the crude *C. majus* extract against *Staphylococcus aureus* (ATTC 6538) (*S. aureus*) and *Pseudomonas aeruginosa* (PA25) (*P. aeruginosa*) was determined according to NCLS M07-A6 guidelines—Methods for Dilution Antimicrobial Susceptibility Tests for Bacteria That Grow Aerobically. The tests were performed using the broth microdilution method on 96 multi-well polystyrene plates (Sigma-Aldrich, (St. Louis, MO, USA)). First, a stock of *C. majus* solutions was prepared in a range from 0.15 to 10 mg/mL in sterile MHB. Then, 50 µL of each *C. majus* dilution was added to microplate wells containing 50 µL of the bacterial suspension, previously adjusted to approximately ~10^7^ CFU/mL in MHB. The assay was performed in triplicate, and the 96-well plates incubated for 24 h at 37 °C. The MIC of *C. majus* was defined as the lowest concentration at which no visible bacterial growth could be detected on the bottom of each well (medium-turbidity not identified).

### 2.4. Emulsion Electrospinning Process

A PCL solution was prepared by dissolving (8% *w*/*v*) PCL in a chloroform/DMF (volume ratio of 30:20) mixture at 50 °C. This solution was kept under magnetic stirring overnight to ensure the proper dissolution of the polymer. To prepare the PVA_PEC blend, 8% PVA (*w*/*v*) and 2% PEC (*w*/*v*) solutions were dissolved in distilled water at 90 °C and at room temperature, respectively. These solutions were first prepared separately and then blended with a 7:3 ratio of PVA and PEC. Then, this blend was further incorporated with 2.5% on weight of fiber (owf) of crude *C. majus* extract. After that, the PVA_PEC blend containing *C. majus* extract was added into the PCL solution with a ratio of 1:1 (*v*/*v*), and the resultant mixture was stirred using a high-speed stirrer Techmatic S2 to produce the water-in-oil (W/O) emulsion. The final blend was stirred at room temperature for 4 h to ensure stable and uniform emulsion. As the control, PCL, PVA_PEC, and plain PCL/PVA_PEC were prepared in the same conditions.

After preparation, the solutions were immediately electrospun using Nanospider Technology (Nanospider laboratory machine NS LAB 500S from Elmarco s.r.o., Liberec, Czech Republic, a needle-free electrospinning apparatus, based on a rotating spinning electrode, and then the electrospun nanofibrous meshes were fabricated. The PCL was electrospun using an applied voltage of 80.0 kV with a working distance (distance from the electrode to collector) of 13 cm and an electrode rotation rate of 55 Hz, while PVA/PEC blend was electrospun applying 75.0 kV, a working distance of 15 cm and an electrode rotation rate of 45 Hz. In the case of emulsions (PCL/PVA_PEC and PCL/PVA_PEC containing crude *C. majus* extract), processing conditions were respectively set at 80 kV, 15 cm, 55 Hz. All solutions were electrospun for 1 h at 25 °C and relative humidity up to 35%. The electrospun nanofibrous meshes were collected on polypropylene non-woven fabric and dried in the hood at room temperature till constant weight.

### 2.5. Characterization of the Produced Electrospun Nanofibrous Meshes

#### 2.5.1. Fourier Transform Infrared Spectroscopy Study

Attenuated total reflectance Fourier transform infrared (ATR-FTIR) spectroscopy (Thermo-Nicolet is10 FT-IR Spectrophotometer, Waltham, MA, USA) was used to analyze the functional groups present in the electrospun nanofibrous meshes produced through emulsion electrospinning. Samples were recorded over the range of 400–4000 cm^−1^ at 32 scans min^−1^ and averaged at the resolution of 4 cm^−1^.

#### 2.5.2. Scanning Electron Microscopy (SEM) Imaging and Analysis

The morphology and surface topography of raw electrospun PCL nanofibers, PVA_PEC blend, and both the plain and the *C. majus*-loaded PCL/PVA_PEC nanofibrous meshes were examined using an SEM (Hitachi S2700, Tokyo, Japan). First, the samples were made electrically conductive by coating with a thin gold layer using an Emitech K550 sputter coater (Emitech Ltd., Ashford, Kent, UK) and then imaged at 5000× magnification. The obtained SEM images were analyzed by an image-processing software (ImageJ, National Institutes of Health, Bethesda, MD, USA) for the determination of the diameters of the fibers. The diameter frequency distributions were further assessed with GraphPad Prism 6 software (Prism Software, La Jolla, CA, USA).

#### 2.5.3. Mechanical Properties Characterization

The mechanical properties of both the plain and the *C. majus*-loaded PCL/PVA_PEC nanofibrous meshes were determined in dry conditions with a universal tensile test machine (DY-35 Adamel Lhomargy, Paris, France), according to the American Society for Testing and Materials (ASTM) standard D3039/D3039M (The standard test method for tensile properties of polymer matrix composite materials) by applying a 10-N load cell at a crosshead speed of 2 mm/min, under ambient conditions. All the samples (*n* = 5) were cut into a rectangular shape with dimensions of 1 cm × 4 cm (width × length), and then were vertically fixed between the two automatic gripping units of the tensile tester, leaving a 1 cm gauge length for mechanical loading. The sample thicknesses were measured using an electronic micrometer (Adamel Lhomargy MI20, Draveil, France). Tensile strength and elongation at break were determined from the tensile stress-strain curves, as well as Young’s modulus. The results were expressed as the mean ± standard deviation (SD).

#### 2.5.4. Porosity Measurements

The porosity of the produced electrospun nanofibrous meshes was measured using a liquid displacement method, as described previously by Yeh et al. [39]. The displacement liquid used in this study was absolute ethanol since it penetrates quickly into the pores of materials without inducing any significant shrinkage or swelling of the polymers. Briefly, a dried sample with a known weight (*W_s_*) was immersed in a graduated cylinder containing a known volume of displacement liquid (*W_1_*) and sonicated at 30 °C in an ultrasonic bath (Ultrasons-H, P-Selecta) for 40 min. The total amount of ethanol in the cylinder and the ethanol-impregnated sample was market as *W_2_*. The ethanol-impregnated sample was removed from the displacement liquid, and the volume of remaining ethanol was recorded as *W_3_*. The porosity (*ε*) of porous electrospun nanofibrous meshes was determined as follows (Equations (1)–(3)):(1)$$V_{p}=\frac{\left(W_{2}-W_{3}-W_{S}\right)}{\rho_{\varepsilon }}$$
(2)$$V_{s}=\frac{\left(W_{1}-\left(W_{2}-W_{S}\right)\right)}{\rho_{\varepsilon }}$$
(3)$$\varepsilon \;\left(\%\right)=\frac{V_{p}}{\left(V_{p}+V_{s}\right)}\times 100\;\Leftrightarrow \varepsilon \;\left(\%\right)=\frac{\left(W_{2}-W_{3}-W_{s}\right)}{\left(W_{1}-W_{3}\right)}\times 100$$
where $V_{p}$ is the volume of the sample pores, $V_{s}$ is the volume of the sample, and $\rho_{\varepsilon }$ is the density of ethanol (g/mL). For each sample, the porosity measurements were performed in triplicate, and the average values reported below.

#### 2.5.5. Analysis of the In Vitro Swelling Behavior

The swelling properties of both the plain and the *C. majus*-loaded PCL/PVA_PEC nanofibrous meshes were investigated using a gravimetric method. The pre-weighed dry samples (*W_dry_*) were immersed in a PBS buffer solution with pH = 5.5 at 37 °C for 30 days. The swollen samples were taken out from the PBS buffer solution at specific time points, and their wet weight measured after gently removing the excess water with a filter paper (*W_wet_*). All measurements were conducted in triplicate, and the swelling ratio calculated according to Equation (4):(4)$$\mathrm{Swelling}\;\mathrm{Ability}\;\left(\%\right)=\frac{\left(W_{wet}-W_{dry}\right)}{W_{dry}}\times 100$$

#### 2.5.6. Study of the In Vitro Degradation Profile

The hydrolytic degradation profile of the plain and the *C. majus*-loaded electrospun PCL/PVA_PEC nanofibrous meshes was evaluated by immersing the samples at 37 °C in PBS solution (pH = 5.5) for different periods up to 30 days. Briefly, at specific time points (1, 4, 8, 16, and 30 days), the samples were taken out from the PBS solution and then were dried and weighed. Finally, the degradation percentage was determined using the following Equation (5):(5)$$\mathrm{Weight}\;\mathrm{loss}\;\left(\%\right)=\frac{W_{i}-W_{t}}{W_{t}}\times 100$$
where $W_{i}$ is the initial weight of the samples, and $W_{t}$ is the weight of the sample at time *t*.

#### 2.5.7. Wettability Studies

The wettability of both the plain and the *C. majus*-loaded PCL/PVA_PEC nanofibrous meshes was measured using a contact angle measurement instrument OCAH-200 (DataPhysics Instruments GmbH, Filderstadt, Germany) operating in static mode at 25 °C. First, samples were placed on the measuring stage. Subsequently, water droplets of size 4 µL were placed at different locations on the sample surfaces from a motorized syringe, and the angle was measured immediately (within 10 s). The mean value was determined from five different points and reported as the contact angle of each sample.

#### 2.5.8. Water Vapor Transmission Rate (WVTR) Analysis

The water vapor transmission rate (WVTR) of both the plain and the *C. majus*-loaded PCL/PVA_PEC nanofibrous meshes was measured according to the ASTM E96/E96M-15 (Standard Test Methods for Water Vapor Transmission of Materials) to evaluate the moisture permeability of the samples. Briefly, each sample was cut into a round shape of 1.2 cm diameter and carefully placed on top of test tubes filled with 10 mL of deionized water at the initial time. Afterward, the samples-glass tubes assembly was placed in an incubator at 37 °C. The weights of the samples were recorded at specific time points, and the weight loss versus time was linearly filled to calculate the WVTR values. WVTR was determined using Equation (6):(6)$$\mathrm{Water}\;\mathrm{vapor}\;\mathrm{transmission}\;\mathrm{rate}\;\left(\mathrm{WVTR}\right)=\frac{W_{loss}}{A}\;\left(g/m^{2}/\mathrm{day}\right)$$
where *W_loss_* is the daily weight loss of water, and *A* is the glass tube open area in m^2^.

### 2.6. Determination of In Vitro Release Profile

The crude *C. majus* extract release was characterized by an in vitro study. The electrospun PCL/PVA_PEC nanofibrous meshes loaded with crude *C. majus* extracts were immersed in PBS pH = 5.5 release medium and stirred at 100 rpm and 37 °C to provide an environment favorable for wound healing. At predetermined time intervals, a fixed volume of release medium was taken out, and an equal amount of fresh PBS refilled. The concentration of crude *C. majus* extract was measured by UV-Vis spectrometry at a wavelength of 360 nm [40]. The absorbance values were converted to release percentages according to the calibrated curve constructed from a series of *C. majus* standard solutions with concentrations from 0.00 mg/mL to 5.00 mg/mL. Finally, the in vitro crude *C. majus* extract release curve was drawn over 30 days. All the measurements were performed in triplicate.

### 2.7. Antibacterial Properties Assessment

The antibacterial properties of the electrospun PCL/PVA_PEC containing crude *C. majus* extracts were evaluated according to ASTM E2180-07 standard (Test Method for Determining the Activity of Incorporated Antimicrobial Agent(s) In Polymeric or Hydrophobic Materials). Briefly, bacterial suspensions of *S. aureus* (ATTC 6538) and *P. aeruginosa* (A25) of ~10^8^ CFU/mL (corresponding to the exponential growth phase) were inoculated into a semi-solid agar slurry prepared by adding 0.30% agar to a 0.85% NaCl solution. Then, a thin layer of the inoculated slurry was spread over on top of each sample. After that, the samples were allowed to gel at room temperature and were analyzed immediately (*T_0h_*) and 18–24 h after incubation (*T_24h_*) at 37 °C. The samples were subjected to vigorous vortex mixing for 1 min in a neutralizing solution to release the agar slurry from the sample. The resultant suspension was serially diluted with saline solution (0.85% NaCl) and plated on growth plates. The agar plates were incubated at 37 °C for 18–24 h, and the number of viable bacterial colonies determined as colony forming units (CFU). For this purpose, viable bacterial colonies were counted after incubation, and CFU mL^−1^ calculated to determine the antibacterial effectiveness. All samples and control were tested in triplicate, and the results presented as mean values of log (CFU) with SD.

### 2.8. In Vitro Cell Viability Assay

The cytotoxicity of both the plain and the *C. majus*-loaded PCL/PVA_PEC nanofibrous meshes and their effect on cellular viability was evaluated with the colorimetric MTT assay according to ISO 10993–5 (Biological evaluation of medical devices—Part 5: Tests for in vitro cytotoxicity). First, the NHDF cells were cultured in medium supplemented with 10% FBS in a humidified incubator at 37 °C under a 5% CO_2_ atmosphere. The culture medium was replenished with fresh media every two days.

On the other hand, the samples cut into round disks were placed at the center of each well in 24-well plates covering 1/10 of their area, and then sterilized by UV irradiation (254 nm, ≈7 mW cm^−2^) for 1 h. After that, 1 × 10^4^ cells/well were seeded in each well containing the sterilized electrospun nanofibrous meshes and incubated with 5% CO_2_ at 37 °C for 1, 3, and 7 days. During these intervals of time, the medium was removed and 1 mL of 0.5 mg/mL MTT reagent solution in fresh culture medium was added to each well and incubated for 4 h under the same conditions. After 4 h, the content of each well was replaced by DMSO to dissolve the formazan crystals. The absorbance of each sample was measured at 570 nm using a xMark^TM^ microplate spectrophotometer (Bio-Rad Laboratories, Inc., Hercules, CA, USA). The cells incubated without samples (*K^−^*) and cells incubated with ethanol (96%) (*K^+^*) were chosen as control groups.

### 2.9. Statistical Analysis

Statistical analysis was performed from the one-way analysis of variance (ANOVA), followed by multiple comparison test Turkey using GraphPad Prism 6 software (GraphPad Software, La Jolla, CA, USA) with a statistical significance of *p* < 0.053.

## 3. Results and Discussion

### 3.1. In Vitro Minimum Inhibitory Concentration (MIC) Test of the Crude Extract of C. majus

The MIC values of crude *C. majus* extract against *S. aureus* and *P. aeruginosa* were found to be 0.625 mg/mL and 1.25 mg/mL, respectively. Zuo et al. [41] reported a MIC value of 1.56 mg/mL for crude *C. majus* extract against *Escherichia coli* ATCC25922 (*E. coli,*
*Gram*-negative bacteria). Such a result agrees with the MIC value of *C. majus* extract obtained in this study against *P. aeruginosa*, a Gram-negative bacterium too. On the other hand, the European Medicines Agency revealed that specific alkaloids extracted from *C.*
*majus*, such as quaternary benzophenanthridine and chelerythrine, exhibited MIC values of 5 µg/mL and 10 µg/mL against *S. aureus* [42]. The results obtained confirmed that MIC values for specific alkaloids obtained from plants are lower than those obtained for the crude extract of the same plant and that the concentration of each phytoconstituent in the plant may differ depending on several factors, such as the plant geographic location, the time of year the plant is harvested, and the chosen extraction method, consequently influencing the obtained MIC values.

### 3.2. Characterization of the Produced Electrospun Nanofibrous Meshes

#### 3.2.1. Fourier Transform Infrared Spectroscopy Study

The ATR-FTIR spectra of the PCL, PVA_PEC, and both the plain PCL/PVA_PEC and *C. majus*-loaded PCL/PVA_PEC nanofibers are shown in Figure 1. All the acquired spectra exhibit the characteristic peaks of PCL near 2890 and 2980 cm^−1^ (symmetric and asymmetric CH_2_ stretching vibration), 1721 cm^−1^ (carbonyl stretching vibration), 1293 cm^−1^ (C-O and C-C stretching vibration), 1240 and 1167 cm^−1^ (symmetric and asymmetric C-O-C stretching vibration) [43]. On the other hand, the spectra of PVA_PEC presents a broad peak around 3320 cm^−1^ (O-H stretching vibration), and bands between 2980 cm^−1^ and 2887 cm^−1^ (C-H stretching vibration), near 1380 cm^−1^ (C-H deformation vibration), 1439 cm^−1^ (CH_2_ deformation vibration), and 1251 cm^−1^ (C-O-C stretching vibration) [35]. Therefore, the spectra of PCL/PVA_PEC and PCL/PVA_PEC containing *C. majus* display the characteristic peaks of both PCL and PVA/PEC, but all the other bands of *C. majus* extract are masked by the vibrations of the functional groups in the PCL/PVA_PEC emulsion blend. However, the intensity of the broad peak around 3300 cm^−1^ increased slightly by the addition of crude *C. majus* extract while no distinct peak shifts were identified. This result suggests that the crude plant extract was well incorporated within the electrospun PCL/PVA_PEC nanofibrous meshes.

#### 3.2.2. Scanning Electron Microscopy (SEM) Imaging and Analysis

The surface morphology and the diameters of the nanofibers produced were assessed through SEM analysis, Figure 2. The PCL (260.56 ± 68.29 nm) and PVA_PEC (208.00 ± 45.46 nm) nanofibers displayed randomly orientated fibers with interconnected pores. On the other hand, the PCL/PVA_PEC and PCL/PVA_PEC containing C. *majus* presented an average fiber diameter of 254.33 ± 64.94 nm and 190.53 ± 56.07 nm, respectively. These results revealed the production of thinner fibers in the presence of *C. majus* extract, due to the reduction of the viscosity of the electrospun solution.

Motealleh et al. [44] also showed a similar effect when chamomile, an herbal drug, was incorporated into electrospun PCL/Polystyrene (PS) (65/35) nanofibers. In this previous study, the authors reported a decrease in the average nanofiber diameters from 268 to 175 nm with chamomile extract incorporation.

Furthermore, such morphological features mimic the collagen fibers present in native ECM (50–500 nm) and can support cell adhesion and proliferation, prevent fluid accumulation, and enhance moisture vapor transmission, which ensures an effective wound healing process [8,13,44].

#### 3.2.3. Mechanical Properties Characterization

The mechanical features of a wound dressing material should be in agreement with the parameters established to the skin’s native structure to prevent displacement of the dressing after it has been applied, as well as the pain during muscular movement [45,46]. To perform this, a synthetic polymer, PCL, which exhibits excellent mechanical strength, was blended with a blend of PVA_PEC and PVA_PEC containing *C. majus,* which display high cell affinity but poor mechanical properties [47]. The tensile strength, Young’s modulus, and elongation at break for these materials were assessed in dry conditions and presented in Table 1.

The electrospun PCL/PVA_PEC nanofibrous meshes showed a value of the Tensile Strength of 3.17 ± 1.18 MPa, whereas the electrospun PCL/PVA_PEC nanofibrous meshes containing *C. majus* exhibited a value of 2.96 ± 0.03 MPa. Moreover, Young’s modulus obtained for PCL/PVA_PEC nanofibers was 17.64 ± 5.30 MPa, whereas for PCL/PVA_PEC nanofibers incorporated with *C. majus* decreased to 15.75 ± 6.46 MPa. The elongation at break assays report that the electrospun PCL/PVA_PEC nanofibrous meshes with and without *C. majus* extract can bear a strain of 17.75 ± 1.34% and 20.50 ± 8.20%, respectively. Hence, these results are in accordance with previously reported values for the mechanical properties of the human skin, demonstrating that the produce electrospun nanofibrous meshes can provide appropriate mechanical support during the healing process [45].

#### 3.2.4. Porosity Measurements

The porosity of a biomaterial for the regeneration of skin has a significant impact on its performance [4,18,48]. The porous structure displayed by the electrospun nanofibrous meshes should allow them to perform gas and fluids exchanges, exudate absorption, and promote cell adhesion, migration, and proliferation, improving a new ECM production during wound healing and regeneration [4,18]. The total porosity of the electrospun nanofibrous meshes is presented in Figure 3a and reveals that PCL/PVA_PEC nanofibers have the lowest porosity (85.95 ± 5.75%), whereas PCL/PVA_PEC containing *C. majus* displayed the highest porosity (94.38 ± 4.08%), due to the higher number of void spaces available between nanofibers with thinner diameters. Such a result is in agreement with the findings reported by Yousefi et al. [48], who demonstrated a similar effect when they incorporated *Lawsonia*
*inermis* (Henna) extracts into Chitosan/Poly(ethylene oxide) (PEO) nanofibrous scaffolds.

Also recently, researchers have reported that porosities above 90% are the most suitable for facilitating skin tissue repair since they can provide support for cell accommodation and migration and confer a functional environment to promote their growth [4,48].

#### 3.2.5. Analysis of the In Vitro Swelling Behavior

The swelling capability of the biomaterials for wound healing applications is fundamental to absorb excessive amounts of exudate, which are produced mainly during the inflammatory phase of healing. The absorption rate of wound exudates can avoid impaired cell adhesion and migration, maceration of the surrounding tissue, as well as bacterial invasion and consequently, wound infections [48]. In this context, the swelling ratio of the produced electrospun nanofibrous meshes was measured in a PBS solution at specific time points, Figure 3b. The obtained results showed that the *C. majus*-loaded electrospun PCL/PVA_PEC nanofibers exhibit a higher swelling ability (~400%) than the plain PCL/PVA_PEC nanofibers (~300%). This result may be attributed to an increase in the number of polar and hydrophilic functional groups, higher surface area, and porosity of the *C. majus*-loaded nanofibers. Therefore, the obtained results showed that the higher swelling capability displayed by electrospun PCL/PVA_PEC nanofibrous meshes containing *C. majus* extract is more proper for effective exudate absorption and able to promote the healing of damaged skin. According to a similar study by Yousefi et al. [48], the *Lawsonia*
*inermis* extract loaded into Chitosan/PEO nanofibers showed the same effect on swelling behavior of the scaffolds produced. The swelling ratios indicated that the Chitosan/PEO nanofibers, both hydrophilic polymers, provided higher hydrophilic character and surface area for water absorption when *Lawsonia*
*inermis* extract was loaded [48].

#### 3.2.6. Study of the In Vitro Degradation Profile

Although there is a wide variety of wound dressings available on the market, most of them still need to be replaced or removed from the wound site, which can lead to scar tissue formation and the risk of bacterial infection can also be increased [4,18,48]. Presently, to minimize this drawback, researchers are developing new biodegradable wound dressing materials as drug carriers to enhance the wound healing rate and skin tissue regeneration.

In this study, the degradation profile of the produced electrospun nanofibrous meshes was analyzed for 30 days in a PBS solution, Figure 3c. The obtained results showed that the *C. majus*-loaded PCL/PVA_PEC nanofibers exhibited a weight loss of 40.22 ± 2.86%, while the plain PCL/PVA_PEC nanofibers only lost 26.96 ± 2.09% of the initial weight. The increase in the degradation rate when *C. majus* extract was incorporated into electrospun PCL/PVA_PEC nanofibrous meshes is mainly due to the lower fiber thickness that provides a higher porosity and consequently improves the contact with PBS solution, accelerating their degradation. A similar in vitro degradation profile was reported by Yousefi et al. [48], when *Lawsonia*
*inermis* extract was added to the Chitosan/PEO blend solution.

#### 3.2.7. Wettability Studies

Material surface wettability is a critical parameter for the application of the electrospun nanofibrous materials in skin tissue engineering [49,50]. In this context, the water contact angle (WCA) determination has been used to characterize the surface wettability of these materials. According to literature WCA values between 40° and 70° are characteristic of moderate hydrophilic materials and are more proper for supporting cell adhesion, spreading, and proliferation, in comparison with very hydrophilic (WCA < 20°) or hydrophobic (WCA > 90°) surfaces [49,50].

Regarding the obtained results, the plain PCL/PVA_PEC nanofibers exhibited a WCA of 73.85 ± 11.21°, while the *C. majus*-loaded PCL/PVA_PEC nanofibers presented WCA value of 60.30 ± 14.99°. In this case, the produced electrospun PCL/PVA_PEC nanofibrous meshes displayed a moderate hydrophilic character. Moreover, the incorporation of the *C. majus* extract led to a lower WCA value, which might be due to the presence of polar functional groups in the crude plant extract, such as esters and hydroxyl groups that confer a more hydrophilic character to the nanofibers [29]. Therefore, the *C. majus*-loaded PCL/PVA_PEC nanofibrous meshes are more suitable to provide a moist wound environment while supporting cell adhesion and proliferation, and hence can be potentially used to improve the healing process.

#### 3.2.8. Water Vapor Transmission Rate (WVTR) Analysis

A wound dressing material should provide a proper moist environment at the wound site, avoid wound dehydration, and exudates accumulation [4,44,49]. Therefore, the wound surface moisture can be controlled using wound dressing materials with different WVTRs.

Herein, the plain PCL/PVA_PEC nanofibers exhibited a WVTR of 1853.04 ± 204.65 g/m^2^/day, while *C. majus*-loaded PCL/PVA_PEC nanofibers displayed a WVTR of 2019.82 ± 151.01 g/m^2^/day. The determined WVTR values showed that the incorporation of *C. majus* extract into the electrospun PCL/PVA_PEC nanofibrous meshes improved their performance.

Previous literature data repeatedly showed that WVTR values in the range of 2000–2500 g/m^2^/day are more suitable to keep the wound moist, as well as to create an ideal healing environment able to avoid fluid accumulation and potential infection [4,51]. On the other hand, a higher WVTR can lead to the wound dehydration, and a lower WVTR may cause an accumulation of exudate at the wound site, resulting in the breakdown of extracellular matrix components or maceration of the wound and surrounding skin, and consequently delaying the healing [4,51].

Regarding the obtained results, the highly porous *C. majus*-loaded PCL/PVA_PEC nanofibrous meshes presented a better capability to create and maintain a moist wound environment during the skin’s regeneration process. This result is in agreement with the data previously described by Vakilian et al. [52], who also reported a better water vapor exchange between the wound and surrounding environment when a smooth structure with interconnected pores was fabricated from *Lawsonia*
*inermis*-loaded Poly-L-lactic acid (PLLA)/Gelatin nanofibers.

### 3.3. Determination of In Vitro Release Profile

During the healing process, the sustained release of the bioactive compounds from electrospun nanofibers to the wound site has been studied to enhance the desirable wound healing properties and skin regeneration [4,50]. In this way, the in vitro release study was carried out to check the capability and performance of electrospun PCL/PVA_PEC nanofibrous meshes to provide a controlled release of the bioactive compounds present in the crude *C. majus* extract.

Herein, the release profile was analyzed, and the amount of *C. majus* released from electrospun nanofibers measured with UV/VIS spectroscopy, Figure 4. The obtained results showed that electrospun PCL/PVA_PEC nanofiber meshes were able to sustain the release of the crude *C. majus* extract for 30 days, which is essential to prevent bacterial infection and maintain an appropriate wound healing environment. Approximately 65.70 ± 4.13% of crude *C. majus* extract was released from electrospun PCL/PVA_PEC nanofibrous meshes during the test period. This result proved that the application of emulsion electrospinning using W/O emulsions could provide an improved drug sustained release profile when hydrophilic bioactive compounds, such as medicinal plant extracts, are incorporated [53].

### 3.4. Antibacterial Properties Assessment

Several therapeutic agents have been incorporated into electrospun nanofibers to avoid bacterial colonization to the wound surface and subsequent infections, which may lead to bacterial biofilms, delaying the healing process [4,54].

Herein, the antibacterial properties of the electrospun PCL/PVA_PEC nanofibrous meshes containing *C. majus* were evaluated against *S. aureus* and *P. aeruginosa*, the most common pathogens found in skin wound infections, to prevent bacterial penetration and disease.

The obtained results showed antimicrobial activity against both bacteria, Figure 5. However, the electrospun PCL/PVA_PEC nanofibrous meshes containing *C. majus* extract exhibit a higher inhibitory effect against *S. aureus* (99.98 ± 4.43%, 3.82 Log reduction) than against *P. aeruginosa* (95.26 ± 5.52%, 1.32 Log reduction), with *p* < 0.01. Furthermore, the results proved the potential of the crude plant extracts, such as *C. majus*, to be used as a source of natural compounds with valuable antimicrobial activity in wound management. Regarding that, the antibacterial effect of the *C. majus* was found to be related to the capability of some of its alkaloids, such as sanguinarine and chelerythrine, to change the permeability of the bacterial membrane and inhibit the synthesis of bacterial DNA [29]. Therefore, the electrospun PCL/PVA/PEC nanofibrous meshes containing *C. majus* could be useful for preventing and treating wound infections.

### 3.5. In Vitro Cell Viability Assay

An ideal wound dressing should be biocompatible and perform its function without compromised the cellular events involved in wound healing. Moreover, these materials are supposed to reproduce the 3D architecture of the native skin’s ECM, which support cell adhesion and proliferation [55].

The in vitro biocompatibility of the produced electrospun PCL/PVA_PEC nanofibrous meshes was evaluated using MTT assay, and their effects on NHDF cells proliferation are shown in Figure 6. The cell viability of both the plain and the *C. majus*-loaded PCL/PVA_PEC nanofibrous meshes mats did not show any cytotoxic effect over 7 days.

According to the guideline for evaluation of in vitro cytotoxicity of medical devices (ISO 10993-5), the biomaterials can be classified as non-cytotoxic and biocompatible when the cell viability is greater than 70%. Hence, it is possible to conclude that the produced electrospun nanofibrous meshes are safe and can be applied as wound dressing without inducing any toxicity.

## 4. Conclusions

The production of polymeric electrospun nanofibers conjugated with natural products, such as medicinal plants, has been regarded as a promising approach to improve the performance and efficiency of the wound dressing displaying antimicrobial activity. In this study, crude *C. majus* extract, well known for improving the bactericidal activity and the healing process, was successfully incorporated into electrospun PCL/PVA_PEC nanofibrous meshes via emulsion electrospinning for the first time.

The produced dressing materials were characterized by their morphological, chemical, physical, and biological features. The manufactured electrospun PCL/PVA_PEC nanofibrous meshes exhibited morphological similarities with native skin’s ECM structure and demonstrated to be able to ensure the maintenance of a moist environment at the wound site. Moreover, the in vitro release assay revealed that the *C. majus* extract incorporated in the nanofiber structure was gradually released to the medium for 30 days testing period.

Moreover, the produced electrospun nanofiber meshes presented mechanical properties similar to those of the human skin and showed the capability to inhibit *S. aureus* and *P. aeruginosa* growth, without compromising cell viability. Therefore, the loading and controlled release of *C. majus* extract using the electrospun PCL/PVA_PEC nanofibrous meshes could be potentially applied to prevent bacterial wound infection and consequently accelerate the healing process. Soon, in vivo studies will be performed to further characterize the performance of these dressing materials for wound care and management, as well as the possible allergic reactions that *C. majus* extract may cause to patients.

## Figures and Tables

**Figure 1 nanomaterials-11-01785-f001:**
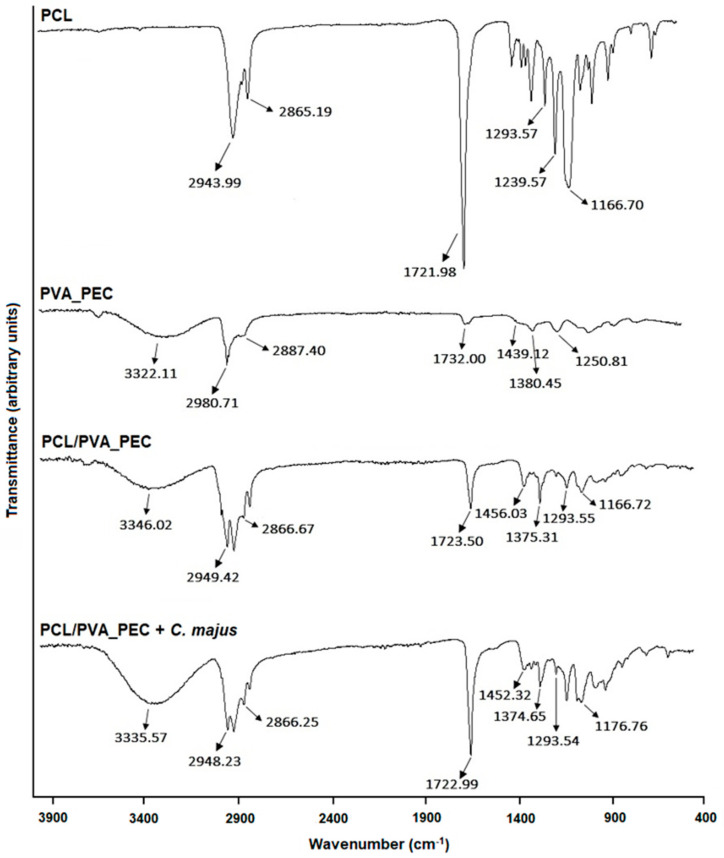
Attenuated total reflection-Fourier transform infrared (ATR-FTIR) spectra of PCL, PVA_PEC, and both the plain and the *C. majus*-loaded electrospun PCL/PVA_PEC nanofibrous meshes.

**Figure 2 nanomaterials-11-01785-f002:**
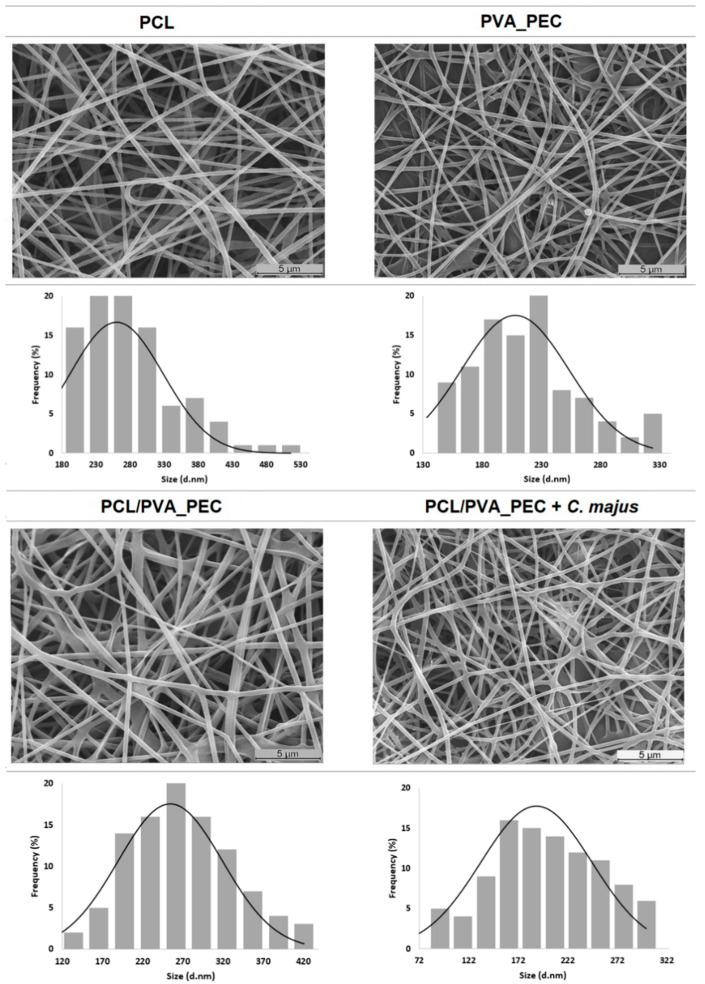
Nanofiber morphology and fiber size distribution of raw electrospun PCL nanofibers, PVA_PEC blend, and both the plain and the *C. majus*-loaded electrospun PCL/PVA_PEC nanofibrous meshes.

**Figure 3 nanomaterials-11-01785-f003:**
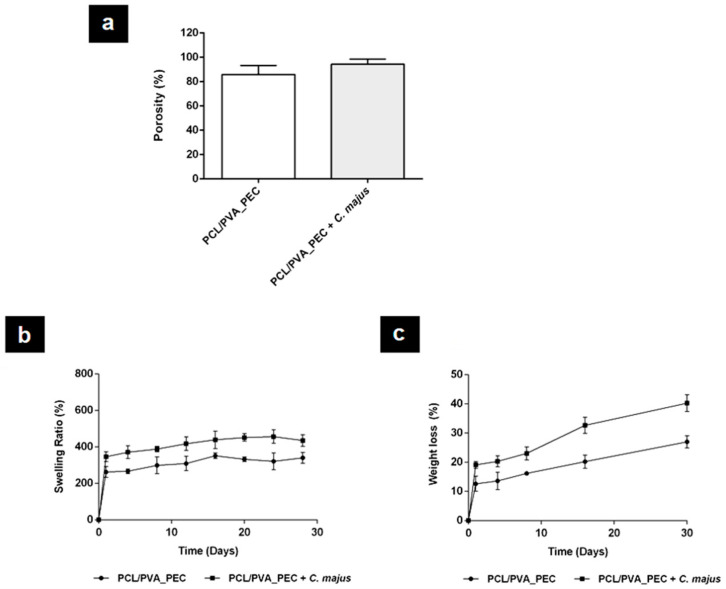
Characterization of the total porosity (**a**), swelling behavior (**b**), and biodegradation profile (weight loss) (**c**) of the produced electrospun nanofibrous meshes.

**Figure 4 nanomaterials-11-01785-f004:**
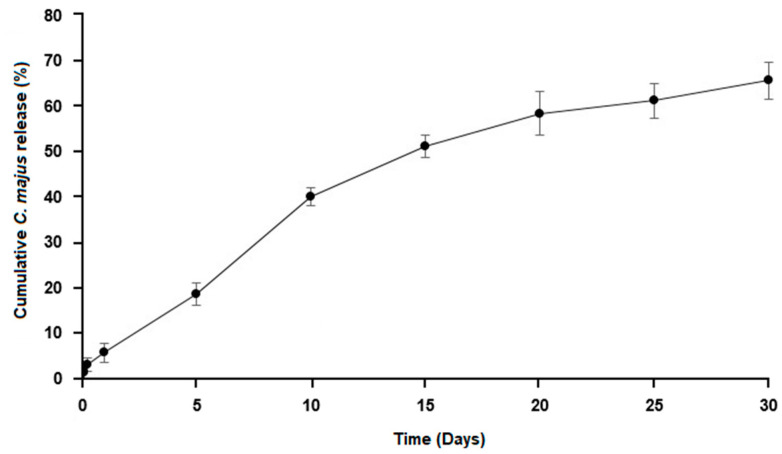
In Vitro release profile of *C. majus* from the produced electrospun PCL/PVA_PEC nanofibrous meshes.

**Figure 5 nanomaterials-11-01785-f005:**
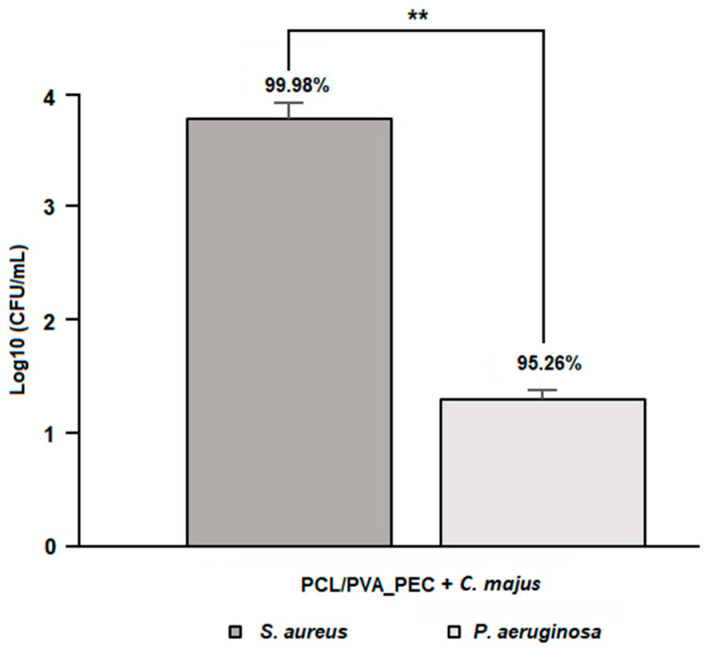
Assessment of antibacterial properties of the produced *C. majus*-loaded electrospun PCL/PVA_PEC nanofibrous meshes against *S. aureus* and *P. aeruginosa*. (Data are presented as the mean ± SD, ** *p* < 0.01).

**Figure 6 nanomaterials-11-01785-f006:**
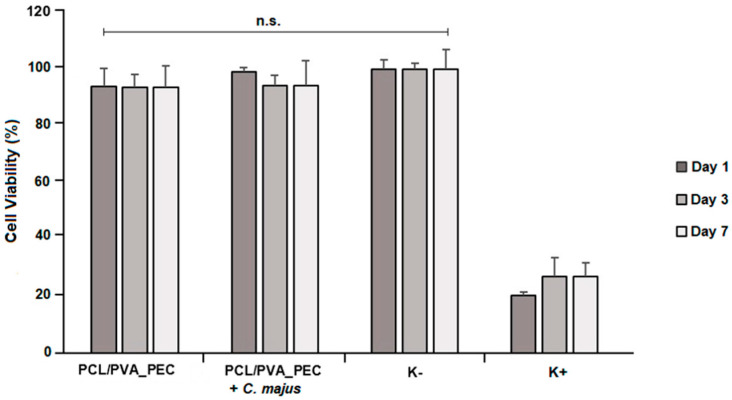
Evaluation of the NHDF cell after 1, 3, and 7 days in direct contact with the produced electrospun PCL/PVA_PEC nanofibrous meshes with and without *C. majus* extract. (“n.s.” indicates not significant (*p* > 0.05)).

**Table 1 nanomaterials-11-01785-t001:** Characterization of the mechanical behavior of the produced PCL/PVA_PEC nanofibrous meshes with and without *C. majus* and comparison with the mechanical features of the native human skin.

	Tensile Strength (MPa)	Young’s Modulus (MPa)	Elongation at Break (%)	Thickness (mm)
PCL/PVA_PEC	3.17 ± 1.18	17.64 ± 5.30	17.75 ± 1.34	0.22 ± 0.01
PCL/PVA_PEC_*C. majus*	2.96 ± 0.03	15.75 ± 6.46	20.50 ± 8.20	0.22 ± 0.03
Native skin	2.50–30.00 ^a^	0.40–20.00 ^a^	10.00–115.00 ^a^	-

^a^ From reference [45].

## Data Availability

The data presented in this study are available in this article.

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
