# Peer review of "Chelidoniummajus L. Incorporated Emulsion Electrospun PCL/PVA_PEC Nanofibrous Meshes for Antibacterial Wound Dressing Applications"

_nanomaterials, 2021, doi:10.3390/nano11071785_

Round 1

Reviewer 1 Report

Electrospinning has emerged as a versatile approach to produce nanofibers useful in drug delivery and wound dressing fabrication. A quite large and recent literature is currently available, documenting the interest and the efforts of the scientific community in this area of research.

The present work was aimed to investigate the preparation of electrospun nanofiber meshes loaded with antimicrobial Chelidonium majus extract and structurally based on a combination of synthetic and natural (bio) polymers such as polycaprolactone (PCL) - polyvinyl alcohol (PVA) - pectin (PEC). Although the basic idea is not particularly innovative, the work is adequately performed with up-to-date methodologies, and experimental data are clearly presented and discussed. Overall, I see this manuscript as a valid proof of concept in an area of research that is probably still in an initial stage of exploration and that will continue to expand, especially for what concern new approaches for development of nanofiber meshes for medical wound dressings, and its practical applications.

Author Response

The authors appreciate and are thankful for your valuable remarks.

Reviewer 2 Report

The authors examine new wound dressing materials which contain extracts from the plant Chelidonium majus L. which were incorporated by a needless emulsion electrospinning technique in Polycaprolactone (PCL)/ Polyvinyl Alcohol (PVA)_Pectin 20 (PEC) nanofibrous meshes. They characterized thorough the physic properties of the new material and test the developed meshes for bacterial inhibition using 2 microorganisms Staphylococcus aureus (S. aureus) and Pseudomonas  aeruginosa (P. aeruginosa) for growth inhibition.

Issues to address:

-The MIC values for the crude C. majus extracts provided,  are very high compared to what they mention for individual compounds from the literature. Information should be given for the concentrations of the bioactive compounds in crude extracts.  Are these variable, or stable? When so much work has been devoted into making and characterizing the new material, it is only appropriate to know the composition of the starting extract.

-in some places in the manuscript, the fonts appear larger L37, L94, L170, L183, L256

- The organisms’ name should be written in italics in all places L19, L27 etc.

-L67 Change to i.e.

-L74 Re-write.

-L92 “Besides, this natural biopolymer can promote” eliminate "besides" or change to Additionally.

-L132 The range of solution concentrations should be written from the smallest to the largest.

-L140 Remove “On the other hand,”

-L176 Change “neat” to plain.

-L191- “without inducing negligible shrinkage or swelling” expression is wrong, do they mean any significant shrinkage or swelling?

-L395 “Currently, to suppress this drawback, researchers are developing new biodegradable wound dressing materials as drug carriers to enhance  the wound healing rate and skin tissue regeneration”. Change to “Presently, to minimize this drawback”.

Author Response

Response to Reviewer 2 Comments

The authors examine new wound dressing materials which contain extracts from the plant Chelidonium majus L. which were incorporated by a needless emulsion electrospinning technique in Polycaprolactone (PCL)/ Polyvinyl Alcohol (PVA)_Pectin (PEC) nanofibrous meshes. They characterized thorough the physic properties of the new material and test the developed meshes for bacterial inhibition using 2 microorganisms Staphylococcus aureus (S. aureus) and Pseudomonas aeruginosa (P. aeruginosa) for growth inhibition.

Issues to address:

Point 1: The MIC values for the crude C. majus extracts provided, are very high compared to what they mention for individual compounds from the literature. Information should be given for the concentrations of the bioactive compounds in crude extracts.  Are these variable, or stable? When so much work has been devoted into making and characterizing the new material, it is only appropriate to know the composition of the starting extract.

Response 1: We agree and appreciate your suggestion, but unfortunately, we can’t prepare the plant extract and quantify the phytochemicals present in the crude ethanol extract of C. majus due to the rules and restrictions on access to the laboratories imposed in response to the COVID-19. Nevertheless, the chemical composition of the crude C. majus extract has been extensively studied and described in the literature by various authors (Page 2, line 87-88). Moreover, it is known that the concentration of each phytoconstituent in the plant may differ depending on several factors, such as the geographic location, the season in which the plant is harvested, and the extraction method chosen, and consequently influence the obtained MIC values. Hence, we have revised the sentence in the revised manuscript (Page 8, line 305-310).

Point 2: in some places in the manuscript, the fonts appear larger L37, L94, L170, L183, L256

Response 2: We accept and agree with the suggestion made. The fonts were unified throughout the revised manuscript.

Point 3: The organisms’ name should be written in italics in all places L19, L27 etc.

Response 3: We apologize and agree with the suggestion made. The organisms’ names were unified in the revised manuscript.

Point 4: L67 Change to i.e.

Response 4: The authors accepted your suggestion and exchanged “I.e.” for “i.e” in the revised manuscript.

Point 5: L74 Re-write.

Response 5: We agree with the suggestion made. The sentence was rewritten accordingly, as the reviewer can check in the revised manuscript (Page 2, line 73-76).

Point 6: L92 “Besides, this natural biopolymer can promote” eliminate "besides" or change to Additionally.

Response 6: The authors accepted your suggestion. We effectively eliminated the word “besides” from the manuscript and replaced it with “Additionally”.

Point 7: L132 The range of solution concentrations should be written from the smallest to the largest.

Response 7: We agree with the suggestion made. The range of solutions concentrations was rewritten accordingly, as the reviewer can check in the revised manuscript.

Point 8: L140 Remove “On the other hand,”

Response 8: The authors agree and accept your suggestion. The sentence was rewritten accordingly, as the reviewer can check in the revised manuscript.

Point 9: L176 Change “neat” to plain.

Response 9: The authors accepted your suggestion and exchanged the word “neat” for the word “plain” in the revised manuscript.

Point 10: L191- “without inducing negligible shrinkage or swelling” expression is wrong, do they mean any significant shrinkage or swelling?

Response 10: The authors agree with your proposal and revised the “without inducing negligible shrinkage or swelling” expression by “without inducing any significant shrinkage or swelling”, as you can check in the revised manuscript (Page 5, line 193-194).

Point 11: L395 “Currently, to suppress this drawback, researchers are developing new biodegradable wound dressing materials as drug carriers to enhance the wound healing rate and skin tissue regeneration”. Change to “Presently, to minimize this drawback”.

Response 11: The authors accepted your suggestion and exchanged “Currently, to suppress this drawback, …” for “Presently, to minimize this drawback, …” in the revised manuscript.

Reviewer 3 Report

The authors presented interesting studies on new dressing materials containing natural products. They used a needleless emulsion electrospinning technique for incorporating Chelidonium majus L. in PCL/PVA_PEC nanofibrous meshes. Moreover, potential use of these electrospun nanofibers as a carrier for Chelidonium majus L. has also been investigated.

Essentially, the experiments have been planned and developed in accordance with the rules, and the results are well, explicitly and correctly presented in the paper. Every observed phenomenon is very well described and discussed and the research results are supported by numerous citations. This is well written as well as an interesting and original manuscript. Moreover, this is an extensive and detailed work. The research results have significant scientific and application value, especially in the medical sector. Generally, I cannot find any essential errors in the article including factual errors. I recommend this manuscript for publication after making a few minor revisions.

Detailed comments (formal comments only):

  1. There should be no space after the "/", e.g. Polycaprolactone (PCL)/ Polyvinyl Alcohol (PVA)/ 81 Pectin (PEC).
  2. Line 150. What is this Cezek Republic country?
  3. Line 170. Different font was used: Emitech K550. It is the same in other chapters. Font should be improved in the article.
  4. Lines 234 and 245: Double space.
  5. Chapter 3 should be moved to the previous page. Chapter 3 should be written directly after the methodology.

Author Response

Response to Reviewer 3 Comments

The authors presented interesting studies on new dressing materials containing natural products. They used a needleless emulsion electrospinning technique for incorporating Chelidonium majus L. in PCL/PVA_PEC nanofibrous meshes. Moreover, potential use of these electrospun nanofibers as a carrier for Chelidonium majus L. has also been investigated.

Essentially, the experiments have been planned and developed in accordance with the rules, and the results are well, explicitly and correctly presented in the paper. Every observed phenomenon is very well described and discussed and the research results are supported by numerous citations. This is well written as well as an interesting and original manuscript. Moreover, this is an extensive and detailed work. The research results have significant scientific and application value, especially in the medical sector. Generally, I cannot find any essential errors in the article including factual errors. I recommend this manuscript for publication after making a few minor revisions.

 Detailed comments (formal comments only):

Point 1 - There should be no space after the "/", e.g. Polycaprolactone (PCL)/ Polyvinyl Alcohol (PVA)/ Pectin (PEC).

Response 1: We agree with the suggestion made. The “Polycaprolactone (PCL)/Polyvinyl Alcohol (PVA)_Pectin (PEC)” was rewritten accordingly, as the reviewer can check in the revised manuscript (Page 1, line 20; Page 2, line 82).

Point 2- Line 150. What is this Cezek Republic country?

Response 2: We apologize for this mistake. The word “Cezek Republic” is misspelled, and it was properly rewritten in the revised manuscript (Page 4, line 152).

Point 3 - Line 170. Different font was used: Emitech K550. It is the same in other chapters. Font should be improved in the article.

Response 3: We accept and agree with the suggestion made. The fonts were unified throughout the revised manuscript.

Point 4 - Lines 234 and 245: Double space.

Response 4: We apologize and agree with the suggestion made. The double space was deleted accordingly.

Point 5 - Chapter 3 should be moved to the previous page. Chapter 3 should be written directly after the methodology.

Response 5: We accept and agree with the suggestion made. The Chapter 3 was moved directly after the methodology, as the reviewer can check in the revised manuscript.

Round 2

Reviewer 2 Report

Issues have been addressed in the revised manuscript.